# A Novel Vegetation Index Approach Using Sentinel-2 Data and Random Forest Algorithm for Estimating Forest Stock Volume in the Helan Mountains, Ningxia, China

Taiyong Ma [1], Yang Hu [2,3,4,5,*], Jie Wang [6], Mukete Beckline [7], Danbo Pang [2,3,4,5], Lin Chen [2,3,4,5], Xilu Ni [2,3,4,5] and Xuebin Li [2,3,4,5]

1 School of Agriculture, Ningxia University, Yinchuan 750021, China
2 Breeding Base for State Key Laboratory of Land Degradation and Ecological Restoration in Northwestern China, Yinchuan 750021, China
3 Key Laboratory of Restoration and Reconstruction of Degraded Ecosystems in Northwestern China of Ministry of Education, Yinchuan 750021, China
4 School of Ecology and Environment, Ningxia University, Yinchuan 750021, China
5 Ningxia Helan Mountain Forest Ecosystem Orientation Observation Research Station, Yinchuan 750021, China
6 College of Grassland Science and Technology, China Agricultural University, Beijing 100093, China
7 Research and Development Unit, Agrosystems Group, Tiko P.O. Box 76, Southwest Region, Cameroon
* Correspondence: huyang@nxu.edu.cn

**Abstract:** Forest stock volume (FSV) is a major indicator of forest ecosystem health and it also plays an important part in understanding the worldwide carbon cycle. A precise comprehension of the distribution patterns and variations of FSV is crucial in the assessment of the sequestration potential of forest carbon and optimization of the management programs of the forest carbon sink. In this study, a novel vegetation index based on Sentinel-2 data for modeling FSV with the random forest (RF) algorithm in Helan Mountains, China has been developed. Among all the other variables and with a correlation coefficient of r = 0.778, the novel vegetation index ($NDVI_{RE}$) developed based on the red-edge bands of the Sentinel-2 data was the most significant. Meanwhile, the model that combined bands and vegetation indices (bands + VIs-based model, BVBM) performed best in the training phase ($R^2 = 0.93$, RMSE = 10.82 m$^3$ha$^{-1}$) and testing phase ($R^2 = 0.60$, RMSE = 27.05 m$^3$ha$^{-1}$). Using the best training model, the FSV of the Helan Mountains was first mapped and an accuracy of 80.46% was obtained. The novel vegetation index developed based on the red-edge bands of the Sentinel-2 data and RF algorithm is thus the most effective method to assess the FSV. In addition, this method can provide a new method to estimate the FSV in other areas, especially in the management of forest carbon sequestration.

**Keywords:** forest stock volume; $NDVI_{RE}$; Sentinel-2; random forest; Helan mountains

## 1. Introduction

Forest stock volume (FSV) refers to the total volume of tree trunks growing within a certain area of a forest, and it is thus an important indicator for measuring the total forest resources within that area [1]. It is also an important parameter to measure forest quality, forest carbon sequestration potentials, and an evaluation of the effectiveness of forest management [2]. Around the globe and ever since the Chinese government formally proposed a strategic plan for carbon peaking and carbon neutrality in 2020, global warming has drawn widespread attention [3–5]. This is because the carbon sink capacity of forests is an effective measure to mitigate global warming. Through the change in FSV [6], the dynamic change in carbon storage can be understood and the carbon sink capacity of the forest ecosystem can be obtained. Therefore, FSV studies are not only paramount in the global carbon cycle, but also practically significant in the realization of China's dual-carbon objectives.

The traditional FSV estimation method is mainly based on the manual measurement of the diameter at breast height (DBH) and tree height on the ground [7]. For fine-scale FSV estimation, it is indeed possible to obtain higher-precision estimation results [8]. However, if extended to a large-scale forest area, the small size and small number of sample plots will make it hard to obtain results close to the actual level [9]. Furthermore, forest ecosystems generally exhibit high spatial heterogeneity and inaccessibility [10,11]. Therefore, at this stage, it is not recommended to estimate FSV purely by manual field surveys. The advent of remote sensing has provided a solution to the challenge of large-scale FSV estimation [1,8,12]. By utilizing satellite images, it is now possible to obtain information about forest structures and compositions across vast areas, without the need for extensive ground measurements [13]. This technology has revolutionized the field of forest inventory, allowing for a more efficient and accurate estimation of FSV at a large-scale. Remote sensing images can be used in combination with a small number of ground samples to obtain highly accurate estimates of FSV or biomass [10]. By calibrating remote sensing data with ground-based measurements, it is possible to create statistical models that can accurately predict FSV at a much larger scale [14]. This combined approach has significant advantages over traditional manual field surveys, as it allows for a more efficient and cost-effective estimation of FSV across large areas. Furthermore, the use of remote sensing data can provide a more comprehensive understanding of forest ecosystems, allowing for more informed management decisions.

However, as more and more optical remote sensing images are applied to FSV studies, researchers have focused on the light saturation phenomenon that affects FSV estimation results [15–17]. Using the band reflectance of optical remote sensing images, all kinds of vegetation indices can be calculated. These traditional indices are usually used to estimate the corresponding FSV or biomass [18–22]. However, as the forest ages, the traditional vegetation indices will no longer respond accordingly to the decrease or increase in tree age [15,16]. This is the phenomenon of overestimation of low values and underestimation of high values that often occurs in FSV estimation studies. This is a result of the insensitivity of spectral variables to changes in FSV, especially in forest areas with high vegetation coverage. Previous studies have explored a variety of methods to decrease the influence of light saturation phenomena on remote sensing estimation. These studies include the utilization of spatial regression models and multi-source remote sensing image fusion [15,17]. Unfortunately, being an FSV study solely on a specific region, it has generalized limitations and it does not apply to other regions.

The present study proposes a novel vegetation index aimed at improving the ability to estimate FSV from remote sensing images. According to the literature, it is known that the Sentinel-2 imagery covers 13 spectral bands [23–26], from visible light to short-wave infrared, and each band has different spatial resolutions. Among all optical satellites, Sentinel-2 is the only satellite that includes three spectral bands in the red-edge range [24,26]. These bands are very effective in monitoring vegetation change information. Such as to estimate the FSV of the Helan Mountains, the vegetation reflectance of these three red-edge bands was used to calculate the novel vegetation index [27]. Similarly, by setting the step size, the optimal weighting coefficient of each red-edge band was determined. As this study was carried out the a typical semi-arid montane forest ecosystem of the Helan Mountains, this study may serve as a knowledge base for related research in similar areas across the globe.

Furthermore, the present study aims at developing a novel vegetation index based on Sentinel-2 multiple red-edge bands. It also combines the original band information and traditional vegetation indices to estimate the FSV of the Helan Mountains under the machine learning algorithm. The study will accomplish the following three goals: (1) to explore the potentials of the novel vegetation index developed based on Sentinel-2 data to estimate the FSV; (2) to compare the ability of the different variable combinations to estimate FSV and determine the best model among the three models developed in this study; (3) to map the FSV distribution of the study area by the best variable combination obtained in objective (2).

## 2. Materials and Methods

### 2.1. Study Area

This study focused on the forest resources in the Helan Mountains National Nature Reserve (38°19′–39°22′ N, 105°49′–106°41′ E) in Ningxia Province (Figure 1). The Helan Mountains belongs to the temperate arid climate zone with typical continental monsoon climate characteristics. The lack of rain and snow all year round leads to a dry climate. Although the average annual temperature is −0.7 °C, there is a wide seasonal variation in precipitation. For instance, the average precipitation from June to September, which accounts for over 62% of the annual precipitation, reaches 260.2 mm. Due to the steep mountain and complex terrain, the Helan Mountains are an important dividing line between climate and vegetation in western China. To the east is the grassland climate and grassland vegetation, and to the west is the desert climate and desert vegetation. It is located at the junction of the Qinghai-Tibetan Plateau, the Mongolian Plateau, and the North China Plain. The special geographical environment has shaped the unique biological groups of the Helan Mountains, making it the only biodiversity hotspot in northern China. Furthermore, the Helan Mountains National Nature Reserve in Ningxia Province has played a major role in studies on the virtuous cycle of vegetation development, succession, and restoration of ecosystems in semi-arid areas.

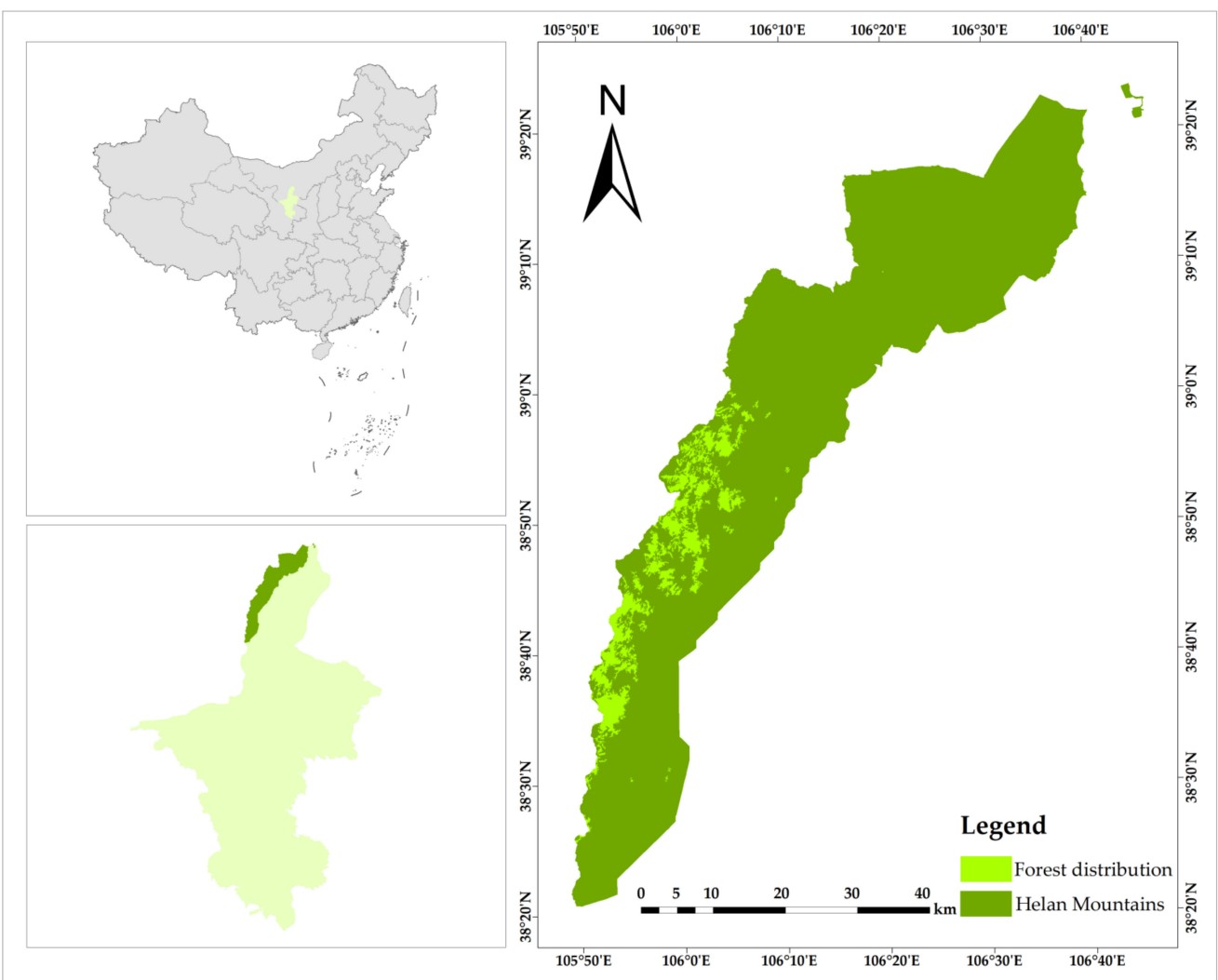

**Figure 1.** The geographical location of the Helan Mountains.

### 2.2. Field Data Collection

The field data were obtained from the 2020 forest resources management "one map" annual update data released by the Ningxia Forestry Survey and Planning Institute. Using these data reduces the workload of field surveys, and it provides access to a large amount of information on ground sample plots. Due to the wide distribution of national surface survey plots, not all sample plots can be surveyed on the spot, and there is a certain degree of uncertainty in these data. Therefore, based on previous studies, the NDVI obtained from Sentinel-2 data was used to screen plots and remove outlier data (NDVI < 0.2) [28,29]. In the end, 881 small class data were extracted for the modeling analysis, and took the hectare stock volume of living trees as the unit area FSV of each sample plot.

Random grouping was used to divide the training data and the testing data. Among the 881 sample plots, 530 (about 60%) were used as the training data and 351 (about 40%) were used as the testing data. Table 1 counts the characteristics of the field FSV training data and testing data, respectively.

**Table 1.** Descriptive statistics of the FSV training and testing data.

| Statistical Category | Training Data ($m^3ha^{-1}$) | Testing Data ($m^3ha^{-1}$) |
|---|---|---|
| Minimum | 3.30 | 6.40 |
| Maximum | 163.20 | 162.30 |
| Median | 45.15 | 48.80 |
| Mean | 56.66 | 63.84 |
| Number of sample plots | 530 | 351 |

### 2.3. The Acquiring and Processing of Sentinel-2 Data

Sentinel-2 covers spectral information in 13 bands, including visible light, near-infrared, red-edge, and short-wave infrared. The Sentinel-2 images are directly extracted from the processed surface reflectance product (COPERNICUS/S2_SR) through the Google Earth Engine (GEE) platform. To match the date of field data and consider the influence of cloud coverage of remote sensing images in the study area, the product date is selected from 1 July 2020 to 31 August 2020. The declouding process uses the method officially announced by the GEE to directly mask out the pixels whose pixel_QA band pixel attributes are 3 and 5. Following cloud removal, the overlaid images were medianized using the median function, followed by coordinate system matching and resampling to 30 m resolution.

#### 2.3.1. Original Band Information

Then, the band information was extracted from the processed images using the vector file of the ground sample. Eight bands (Table 2) of the Sentinel-2 data were selected for this study [30–32], excluding bands 1, 9, 10, 11, and 12 because these bands are mainly associated with the atmosphere or water vapor.

**Table 2.** Selected band information of Sentinel-2.

| Sentinel-2 Bands | Description | Central Wavelength (nm) | Bandwidth (nm) | Resolution (m) | Resampling Resolution (m) |
|---|---|---|---|---|---|
| B2 | Blue | 492.4 | 66 | 10 | 30 |
| B3 | Green | 559.8 | 36 | 10 | 30 |
| B4 | Red | 664.6 | 31 | 10 | 30 |
| B5 | Red Edge 1 | 704.1 | 15 | 20 | 30 |
| B6 | Red Edge 2 | 740.5 | 15 | 20 | 30 |
| B7 | Red Edge 3 | 782.8 | 20 | 20 | 30 |
| B8 | NIR | 832.8 | 106 | 10 | 30 |
| B8A | Narrow NIR | 864.7 | 21 | 20 | 30 |

### 2.3.2. Traditional Vegetation Indices

The potential of six traditional vegetation indices for estimating FSV, calculated from the band reflectance extracted from the Sentinel-2 data (Table 3) were initially tested. Normalized Difference Vegetation Index (NDVI) reflects the background influence of plant canopy and is concerned with vegetation coverage. It is a vegetation index frequently utilized for detecting the growth status of plants. The difference vegetation index (DVI) can also reflect changes in vegetation coverage very well, and within a certain range of vegetation coverage, the DVI rises with the growth of biomass. The ratio vegetation Index (RVI) is a highly sensitive indicator parameter for monitoring green plants, which can be used to detect vegetation status and estimate the FSV. This index is the ratio of light scattered in the near-infrared to light absorbed in the red band, which lessens the effect of the atmosphere and terrain. The perpendicular vegetation index (PVI) represents the vertical distance from the vegetation pixel to the soil brightness line in the two-dimensional coordinate system of R—NIR and is less sensitive to the atmosphere than other vegetation indices. The transformed vegetation index (TVI) is based on the NDVI and introduces a constant of 0.5 to convert the negative value that the NDVI may take into a positive value. The EVI not only inherits the advantages of the NDVI, but also improves the saturation of high vegetation areas, incomplete correction of atmospheric effects, and soil background. The enhanced vegetation index (EVI) can improve the sensitivity of vegetation in high biomass areas and reduce the influence of soil background and atmosphere.

**Table 3.** Several traditional vegetation indices calculated based on Sentinel-2 data.

| Original Vegetation Indices | Formulas | References |
|:---:|:---:|:---:|
| NDVI | $(\rho_{NIR} - \rho_{Red})/(\rho_{NIR} + \rho_{Red})$ | [20] |
| DVI | $\rho_{NIR} - \rho_{Red}$ | [33] |
| RVI | $\rho_{NIR}/\rho_{Red}$ | [34] |
| PVI | $0.939\rho_{NIR} - 0.344\rho_{Red} + 0.9$ | [34] |
| TVI | $\sqrt{(\rho_{NIR} - \rho_{Red})/(\rho_{NIR} + \rho_{Red})} + 0.5$ | [34] |
| EVI | $2.5(\rho_{NIR} - \rho_{Red})/(\rho_{NIR} + 6\rho_{Red} - 7.5\rho_{Blue} + 1)$ | [20] |

### 2.3.3. Novel Vegetation Index Based on Red-Edge Bands

The accuracy of traditional vegetation indices to estimate FSV is severely affected by the light saturation phenomenon. While the three red-edge bands in the Sentinel-2 data have been proven to be an effective way to improve the estimation of the forest parameters, unfortunately only one or two of the red-edge bands were used in existing indices. Therefore, to maximize the ability to estimate FSV using the three red-edge bands in the Sentinel-2 data, a novel vegetation index based on existing NDVI construction principles, the 4-band red-edge NDVI ($NDVI_{RE}$), such as Formula (1) was developed. According to the novel index construction rules, as elaborated in previous studies, in the $NDVI_{RE}$ formula, instead of using the NIR band, the reflectance values of *RE3* and *RE2* are averaged using weights and are substituted. Similarly, the Red band is replaced with a weighted average of the reflectance values of *RE1* and *RE2* [27]. The weighting coefficients "$\alpha$" and "$\beta$" are designed to define the optimal proportion of each band in the construction of the novel index.

$$NDVI_{RE} = \frac{(\alpha \cdot R_{RE3} + (1-\alpha) \cdot R_{RE2}) - (\beta \cdot R_{Red} + (1-\beta) \cdot R_{RE1})}{(\alpha \cdot R_{RE3} + (1-\alpha) \cdot R_{RE2}) + (\beta \cdot R_{Red} + (1-\beta) \cdot R_{RE1})} \tag{1}$$

where $R_{RE1}$, $R_{RE2}$, $R_{RE3}$, and $R_{Red}$ are the reflectance of B5, B6, B7, and B4, respectively. "$\alpha$" and "$\beta$" represent weighting coefficients. The value range of "$\alpha$" and "$\beta$" is (0,1), and the step size is 0.1.

### 2.4. Acquisition of the Forest Distribution Pattern in the Helan Mountains

The Global PALSAR-2/PALSAR Forest/Non-Forest Map product utilizes synthetic aperture radar (SAR) images obtained from the phased array type L-band synthetic aperture radar (PALSAR) on the ALOS-2 satellite to generate a global map of forest and non-forest areas. The classification accuracy of this map, in terms of forest and non-forest information, can reach 90%. This product is widely used for monitoring forest changes, assessing forest carbon storage, and providing information for forest management decisions. We downloaded this product using the GEE and extracted the pixels defined as forest areas within the Helan Mountains region, ultimately obtaining the forest distribution pattern of the Helan Mountains.

### 2.5. Machine Learning Algorithm of Modeling FSV

The random forest (RF) is a machine learning algorithm that uses multiple decision tree classifiers for classification and prediction. In recent years, studies on RF algorithms have rapidly developed accompanied by large numbers of applied research carried out in many fields. The RF algorithm is an efficient bagging-based integrated learning algorithm, and numerous prior studies have shown that the RF algorithm performs well in regression prediction [35–38]. Therefore, this study chooses the RF algorithm for modeling and analysis. The RF algorithm operates by utilizing the bootstrap method, that involves randomly sampling from the original population to create multiple samples. These samples are then used to generate a set of decision trees (ntree). The RF algorithm achieves higher accuracy and robustness by increasing the number of decision trees. At each splitting node, the RF algorithm randomly selects a subset of predictors (mtry) to build each tree. Additionally, there is no need to prune each tree. The RF algorithm employs the "out-of-bag" (OOB) error procedure to independently build each tree based on the training data. This procedure allows for the calculation of variable importance (VI) and OOB error for each tree grown by the RF algorithm. An estimation of the OOB error can be obtained using the following formula:

$$\text{OOB}_{\text{error}} = \frac{1}{n} \sum_{i=1}^{n} (y_i - \hat{y}_i)^2 \qquad (2)$$

where $y_i$ is the measured FSV, $\hat{y}_i$ is the predicted FSV, and $n$ is the total number of OOB samples.

In this study, three RF-based models composed of bands and vegetation indices (VIs) to estimate FSV, namely the bands-based model (BBM), VIs-based model (VBM), and bands + VIs-based model (BVBM) have been used.

### 2.6. Selecting Variables Using the VSURF Package

The VSURF package is a powerful tool for variable selection in regression problems using the RF algorithm. It is a three-step process that involves eliminating irrelevant variables, selecting relevant variables for interpretation, and improving prediction accuracy by removing redundant variables. To begin, the first step of the process involves identifying and eliminating irrelevant variables from the dataset. In the second step, all variables that are associated with the response variable are selected for interpretation. Finally, in the third step, redundant variables are removed to enhance the model's prediction performance. Once the relevant variables have been selected, the minimum mean square error (MSE) is used to determine the optimal number of decision trees (ntree) and the number of variables (mtry) to be used in the RF model. Initially, the ntree parameter is set to 500 and mtry parameter is set to the total number of variables. Once the optimal parameters are calculated, the RF regression model is established and tested.

### 2.7. Assessment of the Modeling Performance

This study utilized two metrics to assess the effectiveness of the RF model. The first metric was the coefficient of determination ($R^2$, Formula (3)), that indicates the extent to which the independent variable can account for the variability in the dependent variable. The second metric was the root mean square error (RMSE, Formula (4)), that represents the

standard deviation of the difference between the observed data and the fitted model. A higher $R^2$ and a lower RMSE are indicative of a well-fitting model. The model is trained on 60% of the total samples, and the remaining 40% are used for testing. This approach allows for accurate predictions while reducing the risk of over-fitting.

$$R^2 = 1 - \frac{\sum_{i=1}^{n}(y_i - \hat{y}_i)^2}{\sum_{i=1}^{n}(y_i - \overline{y})^2} \tag{3}$$

$$\text{RMSE} = \sqrt{\frac{\sum_{i=1}^{n}(y_i - \hat{y}_i)^2}{n}} \tag{4}$$

where $y_i$ is the measured FSV, $\hat{y}_i$ is the predicted FSV, $\overline{y}$ is the mean measured FSV, $i$ is the same index, and $n$ is the number of sample plots.

## 3. Results

### 3.1. Determination of the Optimal Novel Vegetation Index

According to the calculation formula of the novel vegetation index ($NDVI_{RE}$), the value range of the weighting coefficients "$\alpha$" and "$\beta$" is (0,1), and the step size is 0.1, so 121 $NDVI_{RE}$ can be obtained. Python 3.10 software was used to calculate each $NDVI_{RE}$ value of all small class data, and the Pearson correlation coefficient of each $NDVI_{RE}$ with the FSV per unit area was also calculated. Results of the analysis are shown in Figure 2 (correlation is significant at the 0.01 level (two-tailed). In addition, the Pearson correlation coefficient was also put between the traditional NDVI and unit area FSV in the figure for comparison. Results showed the 47th $NDVI_{RE}$ to have the highest correlation coefficient (r = 0.778), which is better than the traditional NDVI (r = 0.767), and its corresponding values of "$\alpha$" and "$\beta$" were 0.4 and 0.2, respectively. Therefore, the optimal $NDVI_{RE}$ was determined and used for the subsequent modeling analysis.

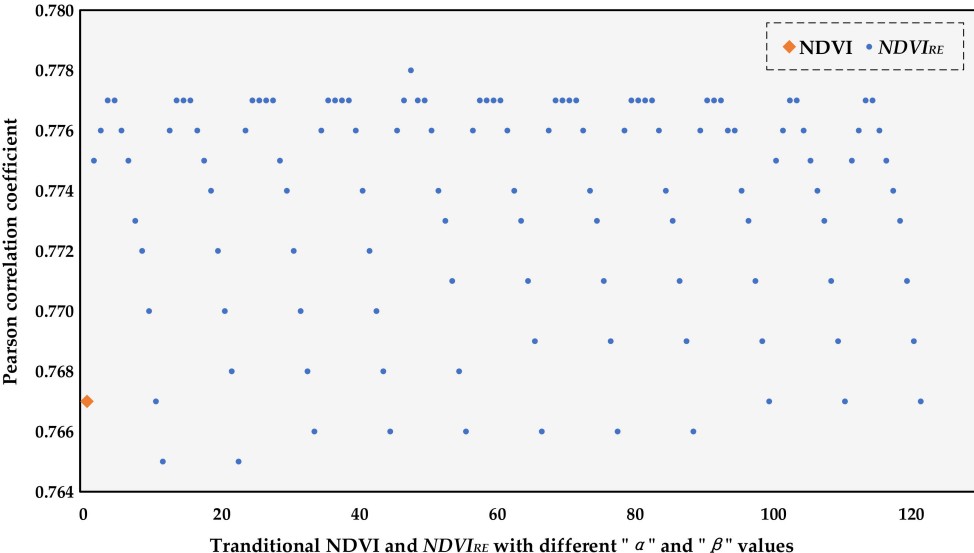

**Figure 2.** Pearson correlation coefficients of the NDVI and $NDVI_{RE}$ with FSV per unit area.

### 3.2. Major Variables Selection and the Importance Related to the FSV Data

Two types of variables, the band (B2, B3, B4, B5, B6, B7, B8, and B8A) and vegetation index (NDVI, DVI, RVI, PVI, TVI, EVI, and $NDVI_{RE}$) were selected to participate in the modeling. Figures 3–5, represent the variable selection process of the three models (BBM, VBM, and BVBM). Meanwhile, Table 4 shows the final variable selection results of each model.

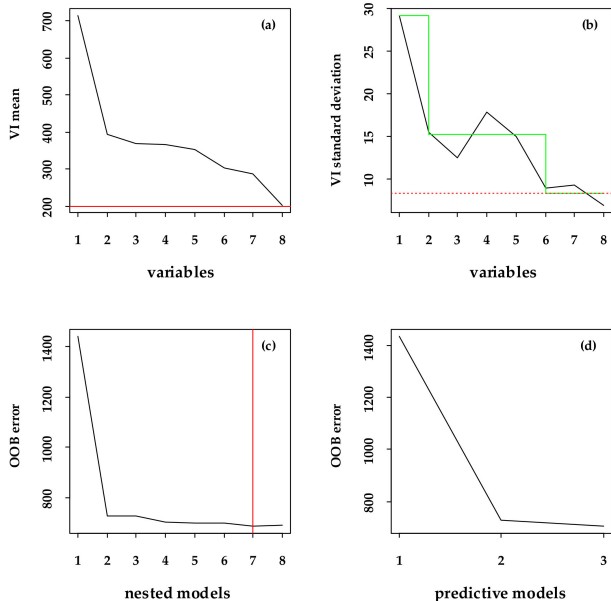

**Figure 3.** The variables selection of BBM. (**a**,**b**) Removes the negatively important variables based on the variable importance (VI) mean and standard deviation, respectively ((**a**), the threshold position is represented by a solid red line that runs horizontally, and (**b**), the green segmented line represents the predicted value given by the CART model, and the red line with dashes running horizontally represents the minimum predicted value). (**c**) Gradually builds a random forest from only the most important variables to all variables selected in the first step, and selects the corresponding variables according to the average OOB error (the vertical solid red line indicates the minimum error position). (**d**) Gives the number of variables meeting the requirements.

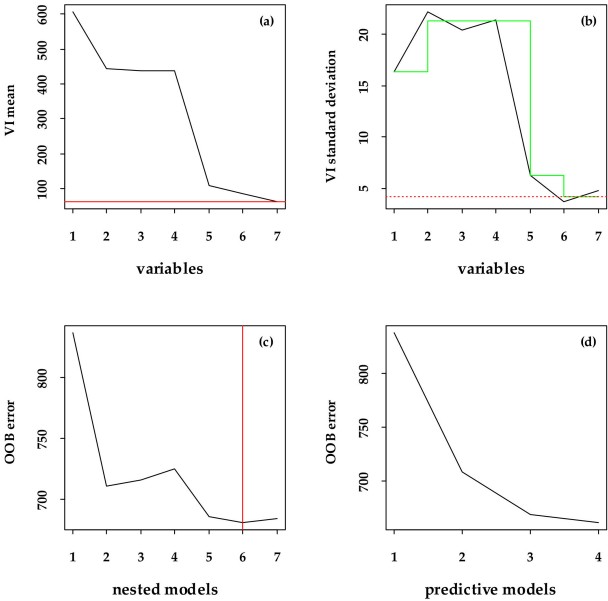

**Figure 4.** The variables selection of VBM. (**a**,**b**) Removes the negatively important variables based on the VI mean and standard deviation, respectively ((**a**), the threshold position is represented by a solid red line that runs horizontally, and (**b**), the green segmented line represents the predicted value given by the CART model, and the red line with dashes running horizontally represents the minimum predicted value). (**c**) Gradually builds a random forest from only the most important variables to all variables selected in the first step, and selects the corresponding variables according to the average OOB error (the vertical solid red line indicates the minimum error position). (**d**) Gives the number of variables meeting the requirements.

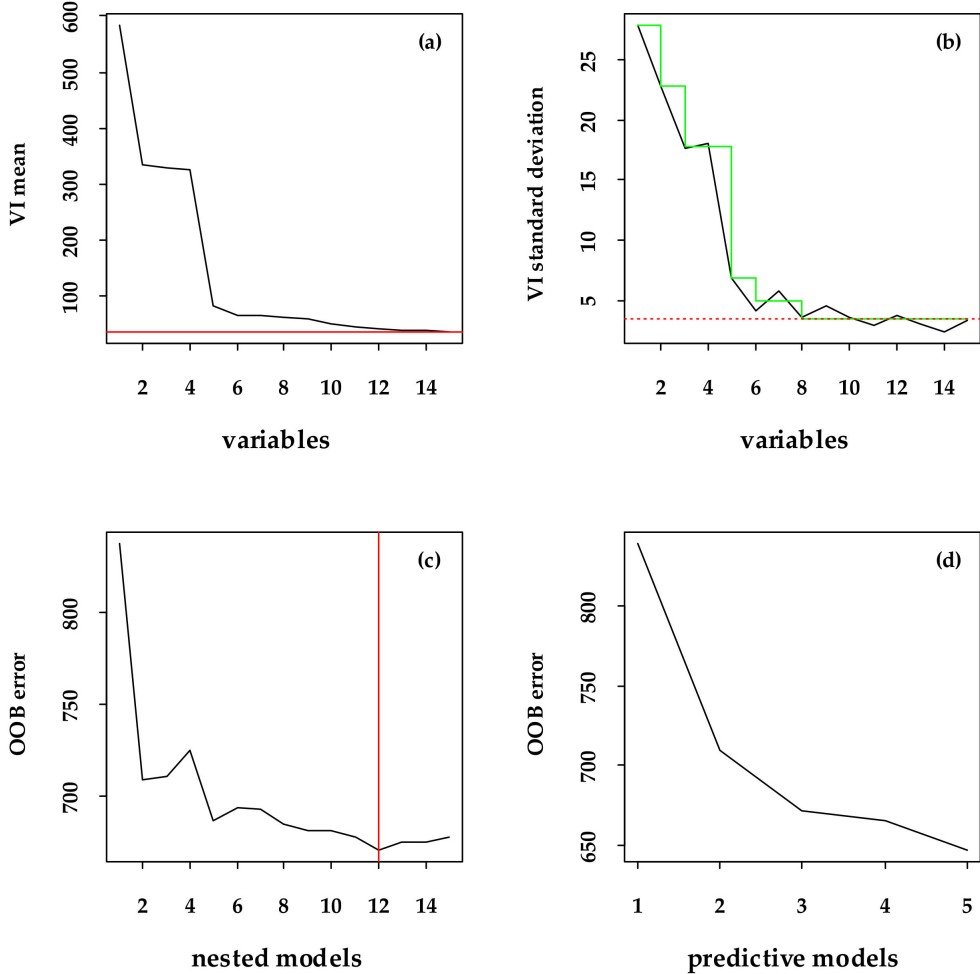

**Figure 5.** The variables selection of BVBM. (**a**,**b**) Removes the negatively important variables based on the VI mean and standard deviation, respectively ((**a**), the threshold position is represented by a solid red line that runs horizontally, and (**b**), the green segmented line represents the predicted value given by the CART model, and the red line with dashes running horizontally represents the minimum predicted value). (**c**) Gradually builds a random forest from only the most important variables to all variables selected in the first step, and selects the corresponding variables according to the average OOB error (the vertical solid red line indicates the minimum error position). (**d**) Gives the number of variables meeting the requirements.

**Table 4.** The variables selection results using the VSURF package.

| RF Models | Variables Selected |
|---|---|
| BBM | B4, B8, B2 |
| VBM | $NDVI_{RE}$, TVI, EVI, DVI |
| BVBM | $NDVI_{RE}$, NDVI, EVI, DVI, B2 |

Furthermore, all predictor variables were ranked based on their ability to estimate FSV using PercentIncMSE and IncNodePurity estimated from the OOB data. The greater the value, the greater the significance of the variable (Figure 6). It is worth noting that the novel vegetation index $NDVI_{RE}$ ranks first in importance under the two evaluation criteria.

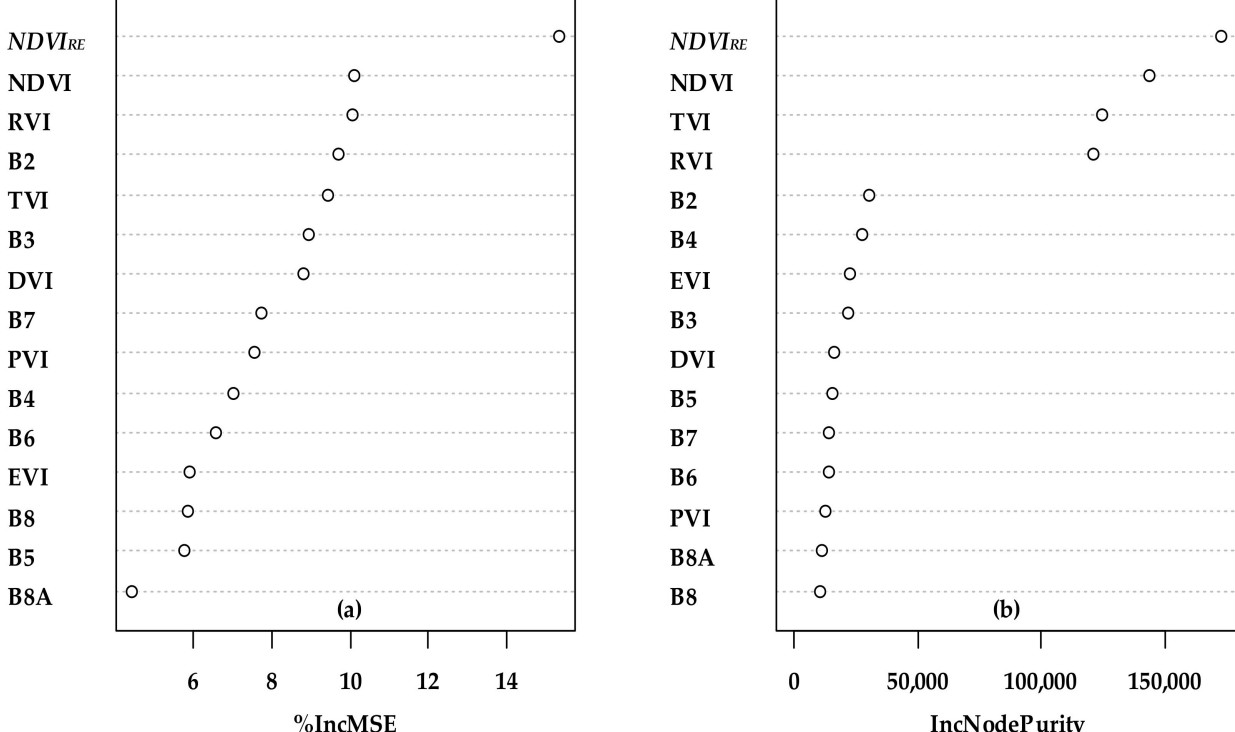

**Figure 6.** Importance ranking plot of all variables. Left, %IncMSE (percentage increase in the mean square error, (**a**)), and right, IncNodePurity (increase in NodePurity, (**b**)).

### 3.3. Optimal Regression Model for the Three Models

To optimize the RF regression model, we need to find the optimal values for two key parameters: "mtry", which determines the number of variables randomly selected as candidates for each split in the decision tree, and "ntree", which determines the total number of trees in the forest that have grown. To calculate the minimum error rate, an iterative algorithm was used, known as an "error rate loop", according to the number of variables participating in the modeling in the three models. Figure 7 shows the determination process of the optimal mtry and ntree of the three models. The values of mtry, ntree, and other performances of each model are summarized in Table 5.

### 3.4. Comparison of the Three Models Predicting FSV

In the training phase, BBM (Figure 8a) with an $R^2 = 0.92$ is slightly better than VBM (Figure 8c) with an $R^2 = 0.91$. However, the RMSE = 11.90 m$^3$ha$^{-1}$ of the VBM is lower than the RMSE = 12.23 m$^3$ha$^{-1}$ of the BBM. The BVBM (Figure 8e) has the highest $R^2 = 0.93$ and the smallest RMSE = 10.82 m$^3$ha$^{-1}$. In the testing phase, the BBM (Figure 8b) with an $R^2 = 0.59$ and RMSE = 27.72 m$^3$ha$^{-1}$ performed almost the same as VBM (Figure 8d) with an $R^2 = 0.59$ and RMSE = 27.32 m$^3$ha$^{-1}$. Similarly, the BVBM (Figure 8f) had the best performance with an $R^2 = 0.60$ and RMSE = 27.05 m$^3$ha$^{-1}$. Obviously, the BVBM is the optimal model in this study, and its predicted FSV is used as the final estimation result to map the FSV. A summary of the data characteristics of FSV as predicted by the three models is presented in Table 6.

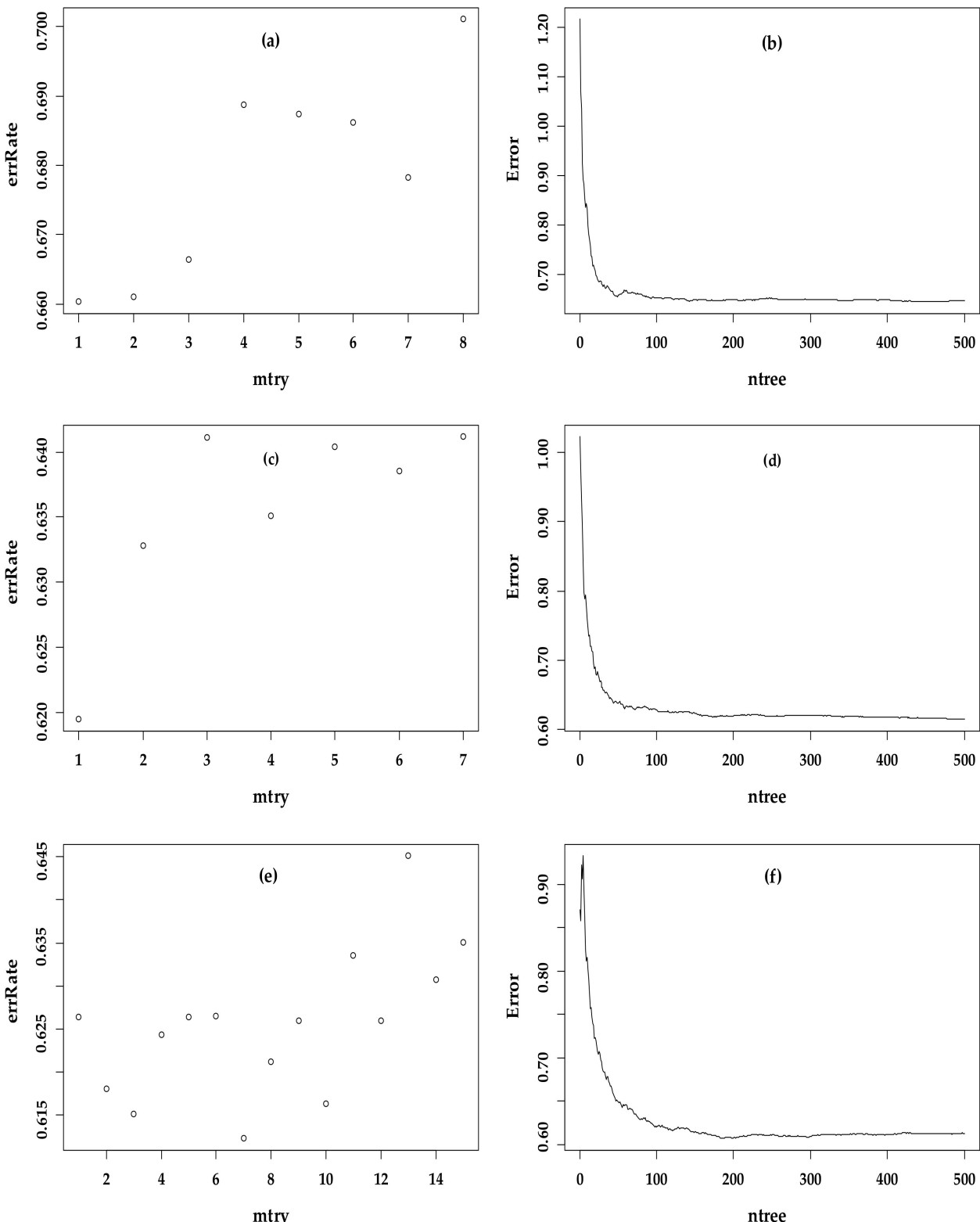

**Figure 7.** (**a**,**c**,**e**) are the distribution of error rate versus mtry; (**b**,**d**,**f**) are the distribution of the error versus ntree; (**a**,**b**) are related to the BBM; (**c**,**d**) are related to the VBM; and (**e**,**f**) are related to the BVBM.

**Table 5.** The best mtry, ntree, and performance of the three models.

| RF Models | mtry | ntree | Mean of Squared Residuals | % Var Explained |
|---|---|---|---|---|
| BBM | 1 | 468 | 636.68 | 56.77 |
| VBM | 1 | 494 | 612.33 | 58.42 |
| BBVM | 7 | 188 | 609.55 | 58.61 |

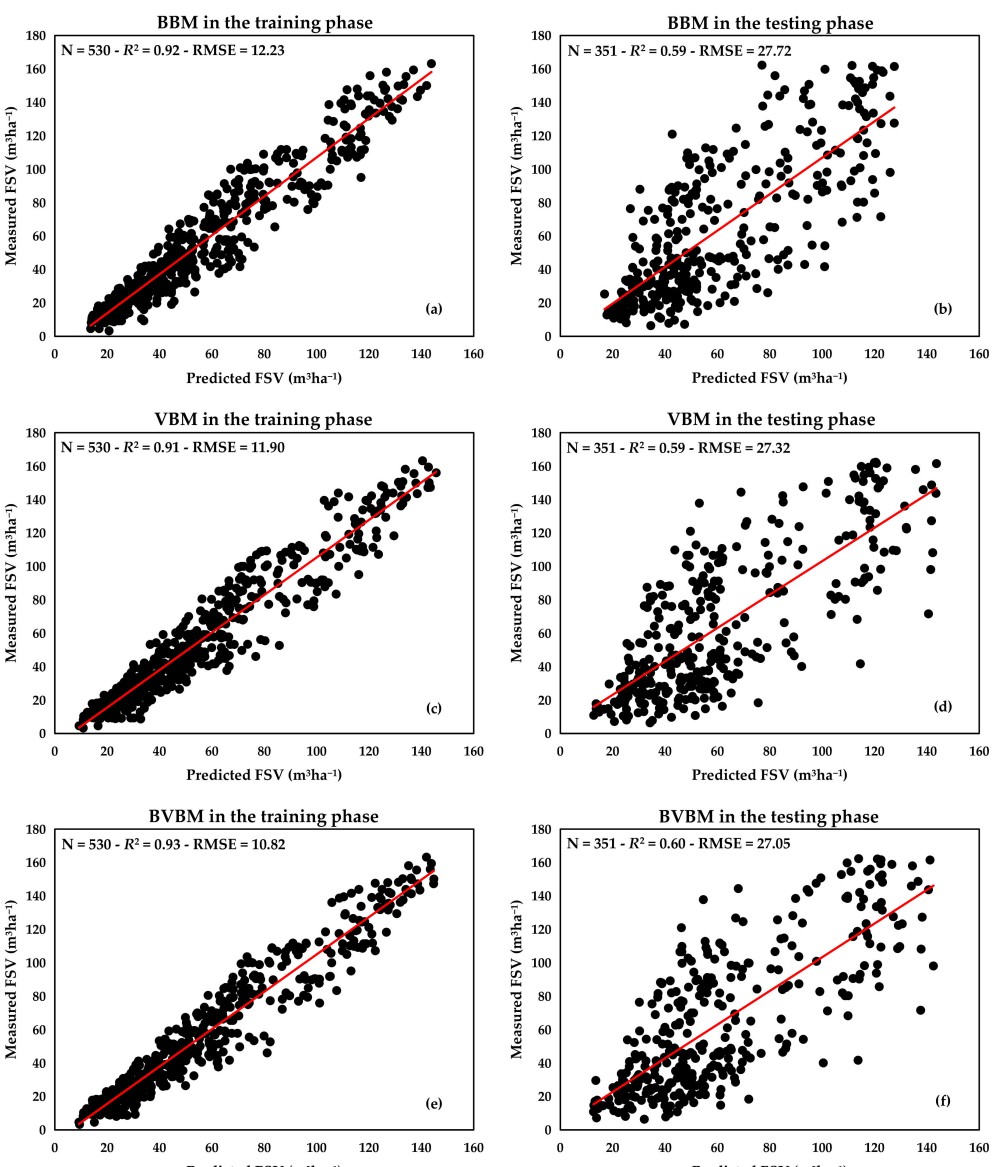

**Figure 8.** Comparison of the measured FSV and predicted FSV by the three models. (**a**), BBM in the training phase. (**b**), BBM in the testing phase. (**c**), VBM in the training phase. (**d**), VBM in the testing phase. (**e**), BVBM in the training phase. (**f**), BVBM in the testing phase.

**Table 6.** Characterization of FSV predicted by the three models.

| Statistical Category | Training Phase (m³ha⁻¹) | | | Testing Phase (m³ha⁻¹) | | |
|---|---|---|---|---|---|---|
| | BBM | VBM | BVBM | BBM | VBM | BVBM |
| Minimum | 13.69 | 9.27 | 9.21 | 16.88 | 12.75 | 12.66 |
| Maximum | 143.83 | 145.66 | 144.76 | 127.55 | 143.64 | 142.48 |
| Median | 48.17 | 47.75 | 47.81 | 50.52 | 52.38 | 50.51 |
| Mean | 56.71 | 56.62 | 56.88 | 60.50 | 60.68 | 60.79 |

*3.5. Mapping FSV Distribution of Helan Mountains*

Based on the results shown in Figure 8, we have concluded that the BVBM is the best-performing model in this study, and we calculated the FSV of the Helan Mountains by the BVBM combined with the forest distribution pattern. Figure 9 is the final FSV map, the minimum value of the unit area FSV of the Helan Mountains is 9.63 m³ha⁻¹ and the maximum value is 143.96 m³ha⁻¹. The total amount of FSV in the Helan Mountains was estimated to be 1,062,727.25 m³. According to the FSV data released by the Helan Mountains National Nature Reserve in Ningxia Province (http://www.hlsbhq.com/, accessed on 22 January 2023), the total FSV of the Helan Mountains is 1,320,721.7 m³. Therefore, the accuracy of the BVBM to predict the FSV in the Helan Mountains reached 80.46%.

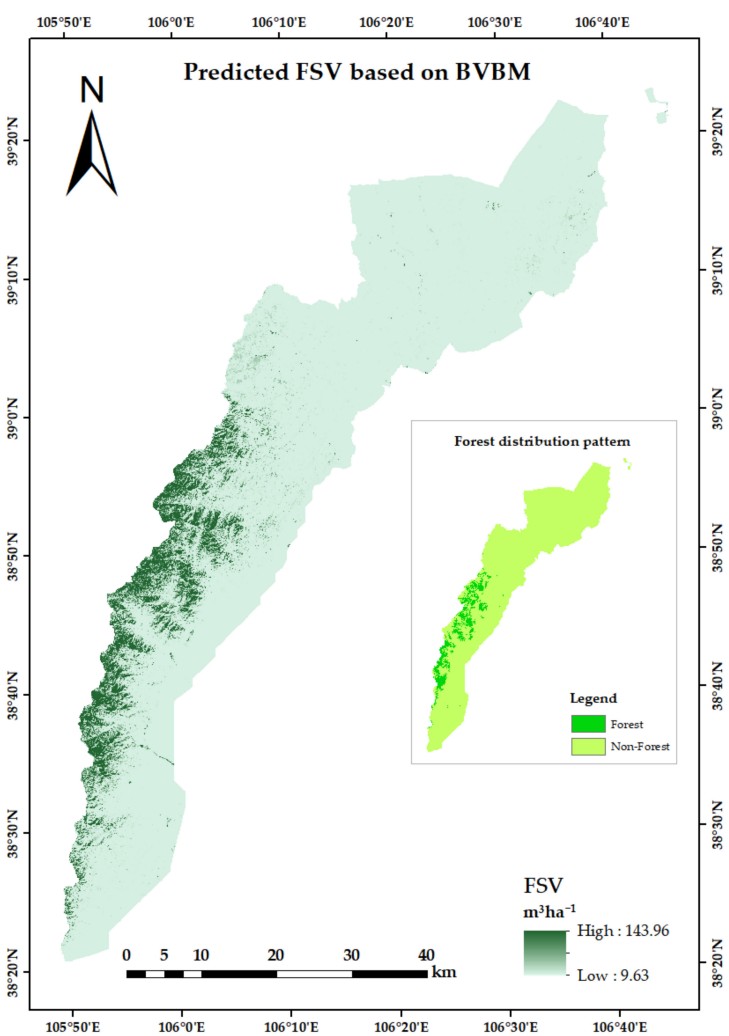

**Figure 9.** Spatial distribution of the predicted FSV, and forest distribution of the Helan Mountains.

## 4. Discussion

The carbon sequestration capacity of montane forest ecosystems is very significant and of prime importance in the global carbon cycle. Due to their geographical location and climatic characteristics, montane forests are an integral part of the entire terrestrial forest ecosystem [35,39]. The Helan Mountains are highly representative of montane forest ecosystems, their FSV estimation has a very high reference value for studies across similar landscapes. However, as a result of the inaccessibility and complex spatial heterogeneity of montane forest ecosystems, it is often a daunting task to obtain a sufficient number and sufficiently representative ground samples to estimate FSV in large-scale areas. Although remote sensing images have made it easier, issues related to low-value overestimation and high-value underestimation still occur [15,17]. However, as more and more red-edge bands in Sentinel-2 data are applied, the accurate estimation of vegetation parameters has been greatly improved [1,2]. For example, based on the red-edge band of Sentinel-2, Liu et al. [27] developed several new vegetation indices to estimate the photosynthetic and non-photosynthetic fractional vegetation cover of alpine grasslands on the Qinghai-Tibetan Plateau. Despite exhibiting a more sensitive response at low vegetation coverage, their study found that compared with traditional vegetation indices, the novel vegetation indices can effectively alleviate the high vegetation saturation problem at low vegetation coverage. In a related study in Zhejiang Province, China, Fang et al. [2] used the optimal variable selection method of different dominant tree species to estimate FSV. Their selected variables included a variety of vegetation indices, such as SRre, MSRre, CIre, and NDI45 developed based on the Sentinel-2 red-edge bands. Almost all of these variables appear in the final variable selection results, which also prove the potential of the red-edge band in estimating forest parameters.

In exploring the potential of $NDVI_{RE}$ to estimate FSV based on the Sentinel-2 red-edge bands, in the variable importance results of the VBM and BVBM, the $NDVI_{RE}$ ranks first. It is worth mentioning that the introduction of weighting coefficients "$\alpha$" and "$\beta$" played a key role in the successful construction of the $NDVI_{RE}$. The results of this study also indicate that the model's estimation accuracy of FSV is significantly improved due to the addition of the $NDVI_{RE}$. First of all, an estimation accuracy of 80.46% is impressive in the research on FSV estimation. Moreover, according to Table 6, we found that the minimum and maximum values in the estimated results of the VBM and BVBM with the $NDVI_{RE}$ involvement are superior to those in the BBM, indicating that the $NDVI_{RE}$ mitigates the issue of light saturation to some extent. In addition, the mean values of FSV predicted by the BVBM in the training phase (56.88 m$^3$ha$^{-1}$) and the testing phase (60.79 m$^3$ha$^{-1}$) are also very close to the mean values of the training data (56.66 m$^3$ha$^{-1}$) and the testing data (63.84 m$^3$ha$^{-1}$).

Despite the proven efficiency and robustness of the RF algorithm through numerous studies [8,21,35–38], there is still a limitation observed in its ability to predict the minimum and maximum values of FSV in both the training and testing phases when compared to the actual training and testing data. This limitation results in overestimation of low values and underestimation of high values. Therefore, it would be necessary for future studies to incorporate more machine learning algorithms and innovative machine learning algorithms. From another perspective, deep learning, as a kind of non-parametric machine learning algorithm, is widely applied in forest monitoring. Numerous prior studies have demonstrated the outstanding capability of deep learning algorithms when it comes to target detection and vegetation classification [40–44].

Another paramount limitation of this study is the source of sample plot data which were the most recent. Although "one map" contains a large amount of necessary forest information, using these data to carry out research can no longer meet the current requirements for real-time forest monitoring. In order to resolve this problem in future studies, it is necessary to use unmanned aerial vehicles (UAVs) to obtain enough measured sample plots. Similarly, many studies have proposed UAVs equipped with hyper-spectral and LiDAR sensors to obtain the horizontal and vertical structure information of forests [45–51]. Its efficiency in obtaining

forest parameters is unmatched by manual investigation. The accuracy of tree height, DBH, and spectral information extracted using UAVs is very close to manual surveys. Therefore, as an innovative research method, it is recommended to use UAVs to replace manual field survey work to improve research efficiency where high-precision forest estimation results can be obtained.

## 5. Conclusions

This study has effectively estimated and mapped the distribution of FSV in the Helan Mountains, with a resolution of 30 m. Utilizing the RF algorithm in conjunction with data from Sentinel-2, the study has affirmed the potential of $NDVI_{RE}$ in FSV estimation. Among all modeled variables, the novel vegetation index $NDVI_{RE}$, constructed based on the three red-edge bands of Sentinel-2, contributed the most to predicting FSV. Furthermore, the BVBM performed the best among the three models based on the two variables of the band and vegetation index. Finally, this study would assist policymakers in designing forest conservation and management paradigms that could potentially support the sustainability and carbon sequestration dynamics in the Helan Mountains and other montane forest ecosystems.

**Author Contributions:** Study design: Y.H. and J.W.; Data curation: T.M.; Investigation, D.P., L.C. and X.N.; Methodology: Y.H. and X.L.; Software, T.M.; Supervision, Y.H.; Writing: T.M.; Writing: review & editing: Y.H. and M.B. All authors have read and agreed to the published version of the manuscript.

**Funding:** This study was supported by the Key Project of Research and Development of Ningxia, China (2021BEB04061, 2022BEG03050), the National Natural Science Foundation of China (32101524), and the National Natural Science Foundation of Ningxia, China (2021AAC03017).

**Data Availability Statement:** The data presented in this study are available upon request from the corresponding author. The data are not publicly available due to funder regulations.

**Conflicts of Interest:** The authors declare no conflict of interest.

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
