# Peer review of "A Novel Vegetation Index Approach Using Sentinel-2 Data and Random Forest Algorithm for Estimating Forest Stock Volume in the Helan Mountains, Ningxia, China"

_remotesensing, doi:10.3390/rs15071853_

Round 1

Reviewer 1 Report

The development of a more accurate method for estimating forest stock volume (FSV) could have significant implications for sustainable forest management and conservation efforts. In this study, the Random Forest algorithm was used to compare the ability of different variable combinations to estimate FSV. The results showed that the newly developed vegetation index, NDVIRE, constructed based on the three Red-Edge bands of Sentinel-2, contributed the most to predicting FSV. Additionally, the BVBM model performed best among the three models based on band and vegetation index variables. The study effectively estimated and mapped the distribution of FSV in the Helan Mountains at a fine scale of 30 meters. To improve the clarity and quality of the paper, the following major revisions are recommended:

General Comments:

1.     Avoid using Chinglish and strive for a more native English expression. This will improve the clarity and readability of the paper.

2.     Provide more context and background information on estimating FSV using remote sensing to support the significance of the study.

3.     Clarify the logic of the paper and ensure that each section flows logically from one to another.

4.     Consider improving the quality and quantity of information provided in the figures and tables.

Specific comments:

1.     The current title is a bit misleading. I recommend changing it to “Estimating Forest Stock Volume in Helan Mountains, China: A Novel Vegetation Index Approach Using Sentinel-2 Data and Random Forest Algorithm”, “A Novel Vegetation Index Approach Using Sentinel-2 Data and Random Forest Algorithm for Estimating Forest Stock Volume in Helan Mountains, China” or something similar.

2.     Instead of "small-scale," use "fine scale" to better describe the resolution of the study.

3.     "Moreso" is not a commonly used word. Consider using "more" or "furthermore" instead.

4.     The second paragraph of the introduction needs more information on the progress of remote sensing for estimating FSV, with references to support the claim. Some of the sentences also need to be revised to make them clearer and less like Chinglish.

5.     The logic of the third and fourth paragraphs is unclear. In the fourth paragraph, there is a bias in estimating FSV using vegetation index, which is still part of the signal saturation problem described in the third paragraph. Clarify the connection between the two paragraphs.

6.     In Figure 1, reduce the map's height and include additional useful information such as administrative boundaries, forest cover, and the location of sampling points. The current map only shows spatial location information, which is not helpful for such a large map. Additionally, adjust the font size of the north arrow to make it more consistent and remove the 0 second coordinate.

7.     A 20-meter resolution is sufficient since only Sentinel-2 data are being used. Spatial resampling is also unnecessary for analysis on the GEE platform and is only required for spatial statistical analysis and data export.

8.     The caption for Table 2 may be misleading, so consider rephrasing it.

9.     Change the caption for Table 3 to "Descriptive statistics of FSV training and testing data." Also, change the row titles to "Maximum," "Minimum," and "Median."

10.  Add a label for the x-axis in Figure 2.

11.  Shorten the caption for Figure 3 to make it clearer. The subplot can be identified as a, b, c, and d. Additionally, consider merging Figures 3, 4, and 5 into a single subplot with a 3 x 4 layout.

12.  In Section 3.4, change "ideal values" to "optimal values."

13.  The first sentence of Section 3.6 needs to be checked for accuracy. The forest area cannot be extracted; only the forest pixels can be extracted. Additionally, the description of FSV in the Helan Mountains is limited and should be expanded.

14.  In the discussion section, it is inaccurate to say that carbon sequestration in the Helan Mountains is critical to the global carbon cycle due to the region's small forest area.

15.  The logic in the second paragraph of the discussion section is confusing. The first part talks about the importance of NDVIRE, while the second part discusses the results of FSV in the Helan Mountains. Clarify the connection between the two.

16.  Avoid excessive focus on limitations in the study compared to its advantages and benefits in the discussion section.

Reviewer 2 Report

This manuscript discussed a novel vegetation index for estimating forest stock volume based on Sentinel-2 data and Random Forest algorithm. Please note the following comments and questions.

1. No line number of the whole manuscript, which is not good for reviewing.

2. In the abstract, both R and R2 were used. It should be unified. Also, the correlation coefficient is usually described as “r” rather than “R”.

3. The word “testing” is much better than “test”. Please revise it.

4. The training obtained the R2 of 0.93, while the testing only obtained 0.60. It seems that the model is not robust.

5. Usually, five keywords are enough. The country name “China” is not often used as one of the keywords. Besides, both abbreviation (NDVIRE) and full name (Forest stock volume) can be seen in the keywords. Please revise it.

6. A blank between “Sentinel-2 data” and “(Table 2)” in 2.3.2 section. Check the whole manuscript.

7. Mostly, researchers use samples in ration of 3:1 or 2:1 for training and testing. However, 60% for training and 40% for testing can be seen in this study.

8. As a research article, there are too many references (65). Among them, more than 30 references are from the same journal “Remote Sensing”. It’s unusual.

9. Besides, there are too many tables and figures in this study. The authors should put important ones in the manuscript. Some are not put in correct section. For example, Table 3 should be in “2. Materials and Methods” section.

10. The English grammars and spelling should be improved. For example, “Moreso, most forest ……”. Please check the whole manuscript carefully.

11. The references format should be unified, such as [29-31], [29, 30].

Reviewer 3 Report

Dear authors,

It was a pleasure to read your manuscript "A Novel Vegetation Index was Proposed to Estimate Forest Stock Volume Based on Sentinel-2 Data and Random Forest Algorithm in Helan Mountains, China". Congratulations!

You have done a large amount of work just from extracting data and spectral bands. Your idea is extraordinary and I hope that the subsequent studies undertaken in the field will validate it.

Honestly, I will try to apply it just to compare the results!

Congratulations once again!

Kind regards,

Round 2

Reviewer 1 Report

Compared with the previous version, the paper has been greatly improved, and the author has made revision based on the most of the comments or suggestions raised last time. The following comments are currently provided to improve the quality of the paper.

1.      Keywords. I recommend to using full name instead of abbreviation for FSV and RF.

2.      A 20-meter resolution is sufficient since only Sentinel-2 data are being used. Spatial resampling is also unnecessary for analysis on the GEE platform and is only required for spatial statistical analysis and data export.

3.      Figure 2. Using different color to plot NDVI and NDVIRE

4.      Shorten the caption for Figure 3 to make it clearer.

5.      The subplot of Figure 6 and Figure 8 can be identified as a, b, c, and d.

6.      Section 3.5. Some of the description belongs to Material and Methods section.

Reviewer 2 Report

The authors answered my comments and questions. The manuscript has the potential to be accepted. 
